# Global-, Regional-, and National-Level Impacts of the COVID-19 Pandemic on Tuberculosis Diagnoses, 2020–2021

**DOI:** 10.3390/microorganisms11092191

**Published:** 2023-08-30

**Authors:** Jorge R. Ledesma, Ann Basting, Huong T. Chu, Jianing Ma, Meixin Zhang, Avina Vongpradith, Amanda Novotney, Jeremy Dalos, Peng Zheng, Christopher J. L. Murray, Hmwe H. Kyu

**Affiliations:** 1Institute for Health Metrics and Evaluation, University of Washington, 3980 15th Ave. NE, Seattle, WA 98195, USA; jledes2@uw.edu (J.R.L.); basting@uw.edu (A.B.); huongc2@uw.edu (H.T.C.); mzhang25@uw.edu (M.Z.); avinav@uw.edu (A.V.); anovo@uw.edu (A.N.); jdalos@uw.edu (J.D.); zhengp@uw.edu (P.Z.); cjlm@uw.edu (C.J.L.M.); 2Department of Epidemiology, Brown University School of Public Health, 121 S Main St, Providence, RI 02912, USA; 3Department of Health Metrics Sciences, University of Washington, 3980 15th Ave. NE, Seattle, WA 98195, USA; 4Center for Biostatistics, Department of Biomedical Informatics, The Ohio State University, 1800 Cannon Drive, Columbus, OH 43210, USA; jianing.ma@osumc.edu

**Keywords:** tuberculosis, diagnosis, notification, COVID-19, SARS-CoV-2, public health measures

## Abstract

Evaluating cross-country variability on the impact of the COVID-19 pandemic on tuberculosis (TB) may provide urgent inputs to control programs as countries recover from the pandemic. We compared expected TB notifications, modeled using trends in annual TB notifications from 2013–2019, with observed TB notifications to compute the observed to expected (OE) ratios for 170 countries. We applied the least absolute shrinkage and selection operator (LASSO) method to identify the covariates, out of 27 pandemic- and tuberculosis-relevant variables, that had the strongest explanatory power for log OE ratios. The COVID-19 pandemic was associated with a 1.55 million (95% CI: 1.26–1.85, 21.0% [17.5–24.6%]) decrease in TB diagnoses in 2020 and a 1.28 million (0.90–1.76, 16.6% [12.1–21.2%]) decrease in 2021 at a global level. India, Indonesia, the Philippines, and China contributed the most to the global declines for both years, while sub-Saharan Africa achieved pre-pandemic levels by 2021 (OE ratio = 1.02 [0.99–1.05]). Age-stratified analyses revealed that the ≥ 65-year-old age group experienced greater relative declines in TB diagnoses compared with the under 65-year-old age group in 2020 (RR = 0.88 [0.81–0.96]) and 2021 (RR = 0.88 [0.79–0.98]) globally. Covariates found to be associated with all-age OE ratios in 2020 were age-standardized smoking prevalence in 2019 (β = 0.973 [0.957–990]), school closures (β = 0.988 [0.977–0.998]), stay-at-home orders (β = 0.993 [0.985–1.00]), SARS-CoV-2 infection rate (β = 0.991 [0.987–0.996]), and proportion of population ≥65 years (β = 0.971 [0.944–0.999]). Further research is needed to clarify the extent to which the observed declines in TB diagnoses were attributable to disruptions in health services, decreases in TB transmission, and COVID-19 mortality among TB patients.

## 1. Introduction

Tuberculosis (TB) is a preventable and treatable communicable disease that remains a major driver of ill health and a global leading infectious cause of death [1]. While global progress in reducing the burden of tuberculosis has improved in recent years [2,3], the emergence of SARS-CoV-2 represented a potential roadblock towards ending the TB epidemic. As a result of the rapid global spread, clinical severity, and capacity to overwhelm health systems of SARS-CoV-2 [4,5], pandemic responses required intensive public health focus and action. Unprecedented pandemic responses, generally in the form of public health and social measures (PHSM) [6], combined with unanticipated surges in care, have resulted in substantial disruptions to various health services across multiple settings [7,8,9], as well as millions of global excess deaths [10,11,12]. Comprehensively investigating the impact of the COVID-19 pandemic on TB cases across geographies is thus of critical significance for tuberculosis programs as countries recover from the pandemic.

Previous studies have documented disruptions across the continuum of care for TB [13], with findings illustrating substantial decreases in TB screening [14,15], up to an extra 35-day delay in TB diagnoses [16,17], reductions in access to TB treatment [18,19], and drops in TB treatment completions [20,21]. Several reasons have been cited for these interruptions, including stay-at-home orders, health services being redirected to COVID-19, reduced hours of health facilities, fear of SARS-CoV-2 infection, and reductions in the ability to pay for care [22,23]. These disruptions have further engendered significant drops in the number of reported TB cases [24,25,26]. Examples include a 63% decrease in India [27], 26% decrease in Kenya [28], and a 15% decrease in Mozambique [29], with some countries reporting unequal reductions by demographic groups in 2020 [30].

However, almost all available studies assessing the impact of COVID-19 on TB notifications were conducted very early in the pandemic, did not provide estimates across countries and regions, and did not assess potential recovery in 2021. The assessment for recovery is particularly important because it may provide indications of the effects of loosening lockdowns and of TB service recovery strategies implemented in countries [31,32]. In addition, estimates across countries provide the opportunity to examine the predictors of disruptions to TB notifications in order to improve our understanding of where and when gaps in TB care exist. While one study conducted in April 2020 observed that no variables that were longitudinally available were correlated with service interruptions for 16 countries [33], there has yet to be a comprehensive analysis of all countries with various cross-country covariates of COVID-19 impact, TB risk factors, COVID-19 PHSM, and socio-demographic factors. 

Moreover, investigating the impact of the COVID-19 pandemic by age, with a particular focus on elderly age groups, may provide further insights on the mechanisms through which TB cases decreased during the pandemic. Researchers have previously theorized that reductions in TB notifications during the pandemic are attributable to interruptions in TB services and to potential reductions in TB transmission owing to pandemic measures (e.g., lockdowns, mask wearing, and reductions in mobility). However, an underexplored area is how the estimated millions of undiagnosed TB patients coming into the COVID-19 pandemic affected TB transmission and outcomes during the pandemic. Previous studies have estimated that 43% of the global population was infected with SARS-CoV-2 at least once during the first two years of the pandemic [34], mortality from COVID-19 is highest among elderly individuals [35,36], and mortality among individuals coinfected with TB and COVID-19 is greater than 20% in LMICs [37]. Considering the widespread nature of the pandemic and the high mortality for coinfected patients, elderly individuals, who also have the highest rates of COVID-19 mortality, that were undiagnosed with TB could be dying from COVID-19. This may further result in reduced TB transmission due to fewer infected individuals in communities and subsequent drops in TB diagnoses. Greater reductions in TB notifications among elderly people may thus be an indicator that individuals in this age group are dying from COVID-19.

The dynamic nature of the COVID-19 pandemic combined with the increasing global availability of TB case data during the pandemic offers the opportunity to extensively evaluate how the COVID-19 pandemic affected TB diagnoses worldwide. In this study, we aimed to quantify the impact of the COVID-19 pandemic on TB diagnoses for 170 countries by comparing the observed TB diagnoses to the expected diagnoses had the pandemic not occurred in 2020 and 2021. We separately quantified the impact of COVID-19 on TB diagnoses for people aged ≥ 65 years old and those under 65 years. Finally, we aimed to identify which covariates, out of 27 pandemic- and tuberculosis-relevant variables, explained the greatest variation in COVID-19 impact on all-age TB diagnoses across countries and globally. These assessments will provide critical insights into the global impact of the pandemic on TB diagnoses and its cross-country variability. 

## 2. Materials and Methods

### 2.1. Data Sources

We utilized country-level data collated by the World Health Organization (WHO) on annual TB notifications between 2013 and 2021 for the 170 countries that reported at least 50 all-age TB cases in 2019. The extracted TB notifications, downloaded on 22 August 2023, were all-forms of TB, including pulmonary TB and extrapulmonary TB, new and relapse TB cases, and TB cases that were bacteriologically confirmed or clinically diagnosed. We separately extracted TB notification data for elderly individuals (aged 65 years and older) and those aged under 65 years. To obtain case notification rates, we used country-level time series of population counts from the Global Burden of Diseases, Injuries, and Risk Factors (GBD) Study [38].

We extracted covariates from multiple well-known databases to explore predictors of cross-country variability on the impact of the COVID-19 pandemic on TB. We first relied on the GBD 2019 study to extract baseline country-level variables on socio-demographic (e.g., income per capita, fraction of population over 65, and education), health-system (e.g., healthcare access and quality index, and hospital beds), and TB risk factors (e.g., smoking prevalence, alcohol consumption, and diabetes prevalence) in 2019 [38,39]. We utilized the Institute for Health Metrics (IHME) COVID-19 modeling database to obtain COVID-19 outcome (e.g., reported cases, deaths, ad total infections) and social distancing measures (e.g., mobility and mask wearing prevalence) in 2020 and 2021 [34,36,40]. Data on COVID-19 PHSM (e.g., stay-at-home orders, school closures) were collated from the Oxford COVID-19 Government Response Tracker (OxCGRT) database [41]. To capture a dose−response effect associated with PHSMs, we measured PHSMs as the number of days with the policy in place for 2020 and 2021 by country.

### 2.2. Statistical Analysis

We first estimated the expected number of all-age TB notifications for the years 2020 and 2021 had the COVID-19 pandemic not occurred by fitting country-specific Quasipoisson regressions with annual TB notifications from 2013 to 2019 as the input. We opted for the quasi-likelihood to account for overdispersion. The regressions in this analysis only included *t* as a count variable, describing the number of years after 2013 with population size as an offset. Mathematically, our country-specific models are denoted as follows:$$log(\lambda_{c}\left(t\right))=\beta_{0}+\beta_{1}t+log(population)$$
where $\lambda_{c}\left(t\right)$ is the number of all-form TB notifications at year t for country c, population as the offset, and $\beta_{0}$ (intercept) and $\beta_{1}$ (linear time trend) are regression parameters. 

We utilized the above country-specific models to predict out counterfactual expected case notifications in the absence of the COVID-19 pandemic for 2020 and 2021. We subsequently linked these expected TB notification estimates with the observed TB notifications to derive the measures of the impact of the COVID-19 pandemic on TB by computing yearly all-age observed-to-expected (OE) ratios. Confidence intervals for OE ratios were computed using a closed form solution that incorporates uncertainty from predictions and observed data points [42]. We rationalized that OE ratios significantly below 1 indicate that the COVID-19 pandemic affected annual TB notifications, which can either occur owing to interruptions to health services, to reductions in transmission, or to both. 

These modeling procedures were repeated for the ≥65-year-old age group and the under 65-year-old age group to obtain age-specific OE ratios. As a result of a lack of consistent reporting by age for some countries, the age-specific analysis was conducted for 155 (of 170) countries. We further compared the likelihood of reduced TB diagnoses because of the COVID-19 pandemic for the ≥65-year-old age group compared to the under 65 age group by computing country-specific relative risks (RR). The OE ratio for the ≥65-year-old age group was the numerator, while the OE ratio for the under 65 age group was the denominator during the computations of RRs. We constructed confidence intervals for RRs using a normal approximation of log age-specific OE ratios. We rationalized that RRs below 1 indicate that the ≥65-year-old group was more impacted by the COVID-19 pandemic compared with the under 65 group, while RRs above 1 indicated that under 65-year-olds were more impacted compared with the ≥65-year-old group. 

We finally examined predictors of cross-country variability in OE ratios for the impact of COVID-19 on the all-age group. We initially tested the association between all-age OE ratios and our set of 27 covariates (Appendix A) using bivariate Pearson *r* correlations and linear regression analyses. For the multivariable analyses, we utilized the Least Absolute Shrinkage and Selection Operator (LASSO) regression procedure to select variables to include in modeling [43]. Briefly, the LASSO procedure is a popular shrinkage method within linear regression models that shrinks the estimates of irrelevant variables towards zero, based on a penalty function, so as to automatically remove the unimportant variables and improve the explanatory power of the model. We determined the value of lambda, the tuning parameter in LASSO, through cross-validation, adaptive and plugin, and minimization of the mean squared error (MSE). After variable selection in LASSO, we inputted the identified variables into a final multivariable linear regression for inferences. We estimated the fraction of variance explained by each covariate in our final multivariable regressions using a Shapley decomposition analysis [44] of $r^{2}$. 

All of the regression analyses examining the association between all-age OE ratios and covariates were stratified by year (2020 and 2021). We also log transformed OE ratios in the analyses to examine the relative impacts of covariates on the outcome. We present both standardized and unstandardized regression coefficients throughout the analysis. Unstandardized coefficients below 1 (and standardized coefficients below 0) suggest that the covariate of interest was associated with an increased difference between the observed and expected all-age TB diagnoses.

## 3. Result

### 3.1. Impact of the COVID-19 Pandemic on TB Diagnoses in 2020

On the global scale in 2020, the world reported 5.83 million TB cases, the fewest since 2013, compared with an expected 7.38 (95% confidence interval [CI]: 7.09–7.68) million TB cases in the absence of the COVID-19 pandemic. This corresponds to a 1.55 (1.26–1.85) million drop in TB cases due to the pandemic and a 21.0% (17.5–24.6%) difference in observed-to-expected TB cases. At the super-region level (Appendix A), the South Asia super-region had the lowest observed-to-expected (OE) ratio had the pandemic not occurred for TB cases in 2020 (OE ratio = 0.73 [0.65–0.81]; difference = −787,000 [−1,060,000–−513,000]), followed by Southeast Asia, East Asia, and Oceania (OE ratio = 0.74 [0.70–0.78]; difference = −601,000 [−706,000–−496,000]); Central Europe, Eastern Europe, and Central Asia (OE ratio = 0.80 [0.78–0.82]; difference = −34,200 [−38,600–−29,800]); Latin American and the Caribbean (OE ratio = 0.82 [0.80–0.84]; difference = −40,000 [−45,000–−35,000]); North Africa and the Middle East (OE ratio = 0.82 [0.80–0.85]; difference = −34,100 [−39,000–−29,100]); High-income (OE ratio = 0.86 [0.85–0.88]; difference = −13,600 [−15,200–−11,900]); and sub-Saharan Africa (OE ratio = 0.97 [0.95–0.99]; difference = −43,900 [−77,200–−10,600]) (Table 1; Appendix A).

Countries that had the largest drops in TB cases due to the COVID-19 pandemic in 2020 included India (608,000 [336,000–879,000]), Indonesia (253,000 [160,000–346,000]), Philippines (192,000 [158,000–227,000]), China (104,000 [74,300–133,000]), and Pakistan (96,200 [58,100–134,000]) (Appendix A). We found that 6 countries had OE ratios below 0.60, 14 countries had ratios between 0.60 and 0.70, and 35 countries had OE ratios between 0.70 and 0.80 (Figure 1A). Among the top 20 high-TB-burden countries in GBD 2019, 14 of them had statistically significant OE ratios below 1 (Appendix A). The Philippines (0.57 [0.51–0.63]), Indonesia (0.60 [0.49–0.72]), India (0.73 [0.62–0.83]), Pakistan (0.74 [0.65–0.83]), and Bangladesh (0.74 [0.71–0.77]) had the smallest OE ratios among the top 20 high-TB-burden countries (Appendix A). However, we observed that 24 countries had OE ratios above 1, 11 of which were in the sub-Saharan Africa super-region, including Nigeria (1.17 [1.06–1.28]), Zambia (1.17, [1.11–1.23]), and Guinea-Bissau (1.12 [1.01–1.23]).

### 3.2. Impact of the COVID-19 Pandemic on TB Diagnoses in 2021

In 2021, the global reported TB cases increased from 5.83 million to 6.43 million. However, there were an expected 7.71 (7.33–8.10) million TB cases in the same year, yielding a 1.28 (0.90–1.67) million decrease in TB cases due to the COVID-19 pandemic, with an OE ratio of 0.83 (0.79–0.88). Every super-region, with the exception of Southeast Asia, East Asia, and Oceania, had larger OE ratios in 2021 compared with in 2020. The South Asia region had the largest increase in OE ratio, from 0.73 (0.65–0.81) to 0.84 (0.74–0.95). In 2021, the Southeast Asia, East Asia, and Oceania super-region contributed the largest decrease in global TB cases as a result of the COVID-19 pandemic, with 721,000 (580,000–863,000), followed by South Asia (493,000 [138,000–849,000]) and Latin America and the Caribbean (29,900 [23,700–36,000]) (Table 1). We found that the COVID-19 pandemic did not affect TB cases in sub-Saharan Africa in 2021, with an OE ratio for the super-region of 1.02 (0.99–1.05).

The same five countries that contributed the most decreases in TB cases during 2020 remained in the top five in 2021, with the exception of Pakistan (38,800 [−8290–85,900]) as it was replaced with Myanmar (69,600 [63,400–75,700]) (Appendix A). However, India (424,000 [72,400–776,000]) and the Philippines (171,000 [125,000–217,000]) contributed slightly fewer cases in 2021 compared with in 2020, while China (126,000 [91,300–161,000]) and Indonesia (279,000 [152,000–406,000]) contributed slightly more cases to the global diminished TB cases as a result of the COVID-19 pandemic. In 2021, 9 countries had OE ratios below 0.60, 8 countries had ratios between 0.60 and 0.70, and 29 countries had OE ratios between 0.70 and 0.80 (Figure 1B). Only 25 countries had OE ratios above 1, with 17 in the Sub-Saharan Africa super-region. However, the number of countries with an insignificant OE ratio increased from 63 in 2020 to 76 in 2021. 

While 12 countries among the top 20 high-TB-burden countries continued to have statistically significant OE ratios below 1 in 2021, 11 of the high-TB-burden countries had higher OE ratios in 2021. Nigeria (1.69 [1.54–1.85]), Bangladesh (0.91 [0.88–0.95]), Pakistan (0.90 [0.78–1.02]), and the Philippines (0.65 [0.58–0.73]) had the largest increases in OE ratios (Appendix A, Appendix A).

### 3.3. Age-Specific Impact of the COVID-19 Pandemic on TB Diagnoses

For the 155 countries with sufficient age-specific data, there were an expected 6.51 (6.12–6.90) million TB cases compared with the observed 5.25 million for the under 65-year-old age group in 2020 (Appendix A). This corresponded to a 1.26 (0.87–1.64) million drop in TB cases due to the COVID-19 pandemic and a 19.3% (14.0–24.7%) difference in observed-to-expected TB cases. Among the ≥65-year-old group in the same year, there were a reported 672,000 TB cases out of an expected 944,000 (901,000–988,000) cases, yielding a 272,000 (229,000–316,000) reduction in TB cases or a 28.8% (25.0–32.7%) decrease. The relative risk (RR) when comparing the ≥65-year-old OE ratio to the under 65 OE ratio was 0.88 (0.81, 0.96) in 2020. Super-regions with the smallest RRs included sub-Saharan Africa (0.82 [0.75–0.90]), Latin America and the Caribbean (0.86 [0.82–0.91]), and South Asia (0.88 [0.74–1.02]). Notable countries with the smallest RRs included South Africa (0.28 [0.23–0.35]), Indonesia (0.73 [0.56–0.97]), and Zimbabwe (0.79 [0.64–0.98]) (Figure 2A, Appendix A).

In 2021, the 155 included countries reported 5.74 million TB cases out of an expected 6.90 (6.37–7.43) million cases, which corresponded to a 1.16 (0.63–1.68) million decrease in TB cases or a 16.8% (9.79–23.7%) drop for under 65-year-olds (Appendix A). For the ≥65-year-olds, 753,000 TB cases were reported compared with an expected 1.02 (0.96–1.09) million TB cases, yielding a 271,000 (207,000–336,000) or a 26.5% (21.1–31.9%) decrease. The global RR in 2021 remained the same as in 2020 at 0.88 (0.79–0.98). Regions with the smallest RRs in 2020 improved in 2021, as the RR for sub-Saharan Africa increased to 0.92 (0.80–1.03), to 0.91 (0.75–1.10) in South Asia, and to 0.95 (0.88–1.01) in Latin America and the Caribbean. At a country level, the RR was lower for Indonesia (0.60 [0.43–0.85]) in 2021, but the RRs improved for South Africa (0.82 [0.69–0.96]) and Zimbabwe (1.05 [0.84–1.30]) (Figure 2B, Appendix A). 

### 3.4. Predictors of COVID-19 Impact on TB Diagnoses

In 2020, 19 of the 27 covariates of interest were associated with all-age OE ratios in the bivariate analyses (Appendix A). The covariates with the strongest effects in 2020 included the reported COVID-19 case rate (standardized β = −0.054 [−0.078–−0.029]; unstandardized β = 0.997 [0.996–0.998]), school closures (standardized β = −0.051 [−0.076–−0.026]; unstandardized β = 0.981 [0.972–0.990]), education (standardized β = −0.047 [−0.072–−0.022]; unstandardized β = 0.968 [0.951–0.985]), indoor air pollution prevalence (standardized β = 0.046 [0.022–0.071]; unstandardized β = 1.007 [1.003–1.010]), and proportion of population ≥65 years old (standardized β = −0.046 [−0.071–−0.021]; unstandardized β = 0.965 [0.947–0.984]) (Figure 3 and Appendix A).

By 2021, only 15 of the 27 focal covariates remained associated with all-age OE ratios (Appendix A). The covariates with the strongest effects in 2021 were age-standardized smoking prevalence (standardized β = −0.071 [−0.100–−0.043]; unstandardized β = 0.957 [0.940–0.974]), school closures (standardized β = −0.049 [−0.078–−0.020]; unstandardized β = 0.991 [0.986–0.996]), the stringency index (standardized β = −0.048 [−0.077–−0.018]; unstandardized β = 0.985 [0.976–0.994]), proportion of population ≥ 65 years old (standardized β = −0.045 [−0.075–−0.016]; unstandardized β = 0.996 [0.944–0.998]), and socio-demographic index (standardized β = −0.043 [−0.073–−0.013]; unstandardized β = 0.988 [0.980–0.996]) (Figure 3 and Appendix A).

The LASSO procedure selected education, age-standardized smoking prevalence, proportion of population ≥65 years old, SARS-CoV-2 infection rate, stay-at-home orders, school closures, reported COVID-19 case rate, and age-standardized reported COVID-19 death rate for all-age OE ratios in 2020 (Table 2). The resulting model explained 28.9% of the variance in OE ratios in 2020, with school closures (17.6%), SARS-CoV-2 infection rate (16.6%), and age-standardized smoking prevalence (16.4%) contributing the most to the explained variance. However, only age-standardized smoking prevalence (unstandardized β = 0.973 [0.957–0.990]), proportion of population ≥ 65 years old (unstandardized β = 0.971 [0.944–0.999]), school closures (unstandardized β = 0.988 [0.977–0.998]), and SARS-CoV-2 infection rate (unstandardized β = 0.991 [0.987–0.996]) remained associated with 2020 OE ratios in this model. 

These results indicate that each 5% increase in the percentage of individuals ≥65 years old was associated with a 2.88% (0.12–5.55%) decrease in all-age OE ratios, each 5% increase in age-standardized smoking prevalence was associated with a 2.68% (0.99–4.33%) decrease, each 30-day period a school closure policy was in place was associated with a 1.24% (0.22–2.26%) decrease, and each 25% increase in SARS-CoV-2 infection rate was associated with a 0.88% (0.42–1.33%) decrease in 2020. Stay-at-home orders trended in the negative direction (unstandardized β = 0.993 [0.985–1.00]), with every 30-day period stay-at-home order associated with a 0.75% (−0.04–1.53%) decrease in all-age OE ratios.

For 2021, the LASSO procedure only selected age-standardized smoking prevalence in 2019 (Table 2). The resulting model explained 12.1% of the variance in 2021 OE ratios (Table 2). The unstandardized regression coefficient for age-standardized smoking prevalence was 0.957 (0.940–0.974), indicating that each 5% increase in smoking prevalence was associated with a 4.30% (2.60–5.97%) decrease in 2021 OE ratios.

## 4. Discussion

This is the first study to comprehensively investigate the impact of the COVID-19 pandemic on tuberculosis (TB) diagnoses for 170 countries across 2020 and 2021. We observed that the pandemic significantly impacted tuberculosis diagnoses by causing a 1.55 million drop in global TB diagnoses in 2020 and a 1.28 million decrease in 2021, with India, Indonesia, the Philippines, and China contributing the most to the global declines. The impact of the COVID-19 pandemic was unequal with respect to age, as we observed that elderly (aged ≥ 65 years old) individuals consistently experienced the largest relative drops in diagnoses compared with the under 65-year-old age group. However, many countries in sub-Saharan Africa were able to reach pre-pandemic levels in TB diagnoses by 2021. Finally, we found that public health and social measures, SARS-CoV-2 infections, and age of the population explained the most variation in cross-country differences between observed and expected TB diagnoses. These findings illustrate that the COVID-19 pandemic had substantial and unequal impacts on TB diagnoses with cross-country variability in recovery to pre-pandemic levels. 

Our results are thus in line with other studies finding that COVID-19-related disruptions were associated with decreases in TB diagnoses [13]. However, it remains unclear how much the decrease in tuberculosis notifications found in this analysis was due to previously described disruptions to TB health services versus potential decreases in TB transmission. Prior research has shown that pandemic responses, in the form of mask wearing, social distancing, and diminished mobility, may reduce transmission, and these measures have been shown to reduce the transmission of other infectious diseases [45,46], including influenza [47]. One study in South Africa suggested that face masks decreases TB transmission by over 50% compared with when patients do not wear face masks [48]. Countries with increased levels of mask wearing in response to the pandemic may thus have experienced protective effects regarding TB transmission. There may be further benefits owing to public health and social measures (PHSM), along with reductions in mobility, as previous work has shown that over 80% of all TB transmissions occur outside of the household for both children and adults [49,50,51]. This may lead to additional reductions in TB transmissions due to fewer community contacts. However, this benefit may be offset by increased opportunities for household transmission owing to prolonged exposure to household contacts and potential increased duration of infectiousness due to disruptions in care [52]. 

Our secondary analysis evaluating the impact of covariates of COVID-19 on TB diagnoses also indicated that both disruptions in TB health services and changes in TB transmission may be the mechanisms through which TB diagnoses decreased during the pandemic. For example, we found that stay-at-home orders and school closures were associated with reduced TB diagnoses during the pandemic. These mitigation measures may have resulted in fewer TB diagnoses owing to either individuals having fewer community contacts and thus fewer opportunities for transmission, and/or to travel restrictions preventing individuals from accessing TB health services. 

The finding that the fraction of population ≥65 years old was selected in covariate procedures, combined with the finding that elderly people experienced the largest relative reductions in TB diagnoses, lends some evidence to the possibility that elderly individuals who were undiagnosed with TB coming into the pandemic died from COVID-19. Several studies have shown high fatality rates among people coinfected with TB and COVID-19, particularly among elderly individuals, and that TB should be considered as risk factor for severe COVID disease and mortality [37,53,54,55]. The risk of mortality may further compound when additional comorbidities are present, such as smoking, which may explain why smoking prevalence was persistently selected in covariate procedures. If elderly individuals, both with undiagnosed and diagnosed TB, died from COVID-19, this might have reduced TB transmission due to fewer individuals who could transmit the disease. Emerging evidence from India’s new national TB prevalence survey [56] covering the period from 2019 to 2021 provides some initial support for potential reduced transmission. The survey found a marked decline in TB prevalence that coincided with the peak in COVID-19 deaths during the second wave in the country, which may indicate that people coinfected with TB and COVID-19 were dying from COVID-19, leading to reduced TB prevalence. This phenomenon has further implications for the many modeling studies concluding thousands of excess TB deaths in the coming years, as these models are dependent on the assumption that any observed reductions in TB diagnoses are a result of health service disruptions. These assumptions may need further validation if additional empirical data show the potential of TB patients dying from COVID-19 and subsequent reductions in the infectious population. In fact, recent longitudinal analyses of cause of death data have shown reductions in undiagnosed TB deaths in South Africa during the pandemic [57], a 50% drop in infectious disease deaths (which included TB deaths) in an Indian city [58], and gradual declines in TB mortality during the pandemic years in Taiwan [59]. It is noteworthy that disruptions in health services may still be a mechanism through which elderly individuals had larger relative reductions, as this population may have delayed access to care due to fears of SARS-CoV-2 infection [22], engendering missed opportunities for TB diagnoses. If they chose to access care, this might be amplified if they could not access TB services due to disruptions. However, if the declines in TB cases were attributable to disruptions in services, we would have observed similar reductions across age groups; a much larger relative decline in the 65 and older age group indicated that there might be additional mechanisms through which TB cases dropped.

Additional data are thus urgently needed so as to disentangle the mechanism through which the COVID-19 pandemic reduced TB diagnoses. However, the current literature has primarily focused on TB health service disruptions. For example, in the case of both India [60] and Indonesia [61], studies have shown that a potential contributor to diminished diagnoses is that health centers temporarily stopped or reduced TB health services in order to focus more on COVID-19 services. For China, strict internal movement restrictions and reduced access to TB diagnostic equipment were cited as drivers for the drops in TB diagnoses among patients [62]. In the case of the Philippines, the emergence of the pandemic led to a discontinuation of active TB case-finding programs and reductions in demands for tuberculosis services [63,64]. Although these studies and others have shown substantial disruptions to tuberculosis health services, a recent review of the impact of COVID-19 on tuberculosis indicated that there are limited empirical data on the effects of the pandemic on tuberculosis transmission [13]. Continued emerging data with increasing availability of cause of death data will provide a better understanding of the impact of COVID-19 on TB and inform proposed mechanisms.

Despite the uncertainties regarding the mechanisms through which the COVID-19 pandemic may have affected TB diagnoses, our study revealed that out of all regions, the sub-Saharan Africa region experienced the fewest drops in TB diagnoses and was the only region to achieve pre-pandemic levels by 2021. This may point to the success of many countries in the region to regularly maintain TB health services during the pandemic. For example, Zambia implemented a multicomponent strategy to improve tuberculosis surveillance in June 2020 that included intensifying active case finding, novel communication campaigns, and weekly stakeholder meetings [65]. Nigeria collaborated early in the pandemic with the private sector, allowing private facilities to uninterruptedly provide care for individuals with TB who were otherwise unable to seek care in the public sector [66]. Notably, sub-Saharan Africa was the region that had the fewest PHSM compared with all of the other regions [41], indicating that the region may not have experienced as many reductions in TB transmission from community contacts compared with other regions. While not in sub-Saharan Africa, Pakistan merits highlighting, considering that it contributed the fifth highest reduction in TB diagnosis in 2020, but recovered and achieved pre-pandemic levels in 2021. This may be a result of the successful implementation of telehealth approaches, increase in mobile TB diagnostic services, and collaborations with the private sector [67]. Notably, the achievement of pre-pandemic levels was unequal in sub-Saharan Africa as the elderly age group had still not reached pre-pandemic levels in 2021, while the under 65-year-old age group did. This may further point to elderly individuals who are both undiagnosed and diagnosed with TB potentially dying from COVID-19.

### Strengths and Limitations

This study has several strengths including the ability to describe the impact of the COVID-19 pandemic on TB diagnoses for over 170 countries across two years of the pandemic. We were also able to collate a comprehensive database of over 25 unique pandemic- and tuberculosis-relevant covariates with sophisticated covariate selection algorithms to identify sets of variables that explained the most cross-country variability in COVID-19 impact on TB. 

However, this study has several limitations. First, our analysis could not distinguish the degree to which reductions in TB transmission and disruptions to health services independently contributed to the differences between observed and expected TB diagnoses. Future analyses will benefit from high-quality TB surveillance data and emerging cause of death data. Second, we could not include other important covariates characterizing degrees of healthcare disruptions, healthcare resiliency, and social mixing/contact into our analysis due to a lack of robust global data. The inclusion of these variables may increase the predictive power of our models as our final regressions only accounted for 29% of the variability in our metrics for the impact of COVID-19 on TB diagnoses in 2020 and only one variable was selected in the 2021 model. Third, our computations of expected TB diagnoses assumed that country-specific historical trends in TB reporting would continue had the pandemic not occurred. Fourth, more detailed TB case data, including disaggregation by subnational, urbanicity, and socioeconomic levels, will improve our understanding of where gaps in diagnoses were the largest and allow for targeted policies to address disruptions. Fifth, there is potential measurement error in some of the included PHSM covariates, as individual metropolitan areas or subnational locations may have deviated from national-level policies according to local infection and behavior levels. Finally, this is an ecologic study and we thus cannot make inferences at an individual level. 

## 5. Conclusions

In this analysis, we observed that approximately 1.55 and 1.28 million fewer TB diagnoses were reported globally in 2020 and 2021, respectively, than what was expected had the COVID-19 pandemic not occurred. We identified substantial cross-country differences in TB diagnosis decreases, but sub-Saharan Africa was the only region that achieved pre-pandemic levels in 2021. In both years, the pandemic had unequal impacts as elderly individuals aged 65 years and older experienced substantially greater relative declines in cases compared with the under 65 age group. Stay-at-home orders, school closures, SARS-CoV-2 infection rates, smoking prevalence, and aging of the population were associated with an increased difference between observed and expected TB diagnoses Additional research is needed to clarify the degree to which the effects of these variables on TB diagnoses were independently and jointly attributable to disruptions in TB health services, potential decreases in TB transmission, and COVID-19 mortality among TB patients.

## Figures and Tables

**Figure 1 microorganisms-11-02191-f001:**
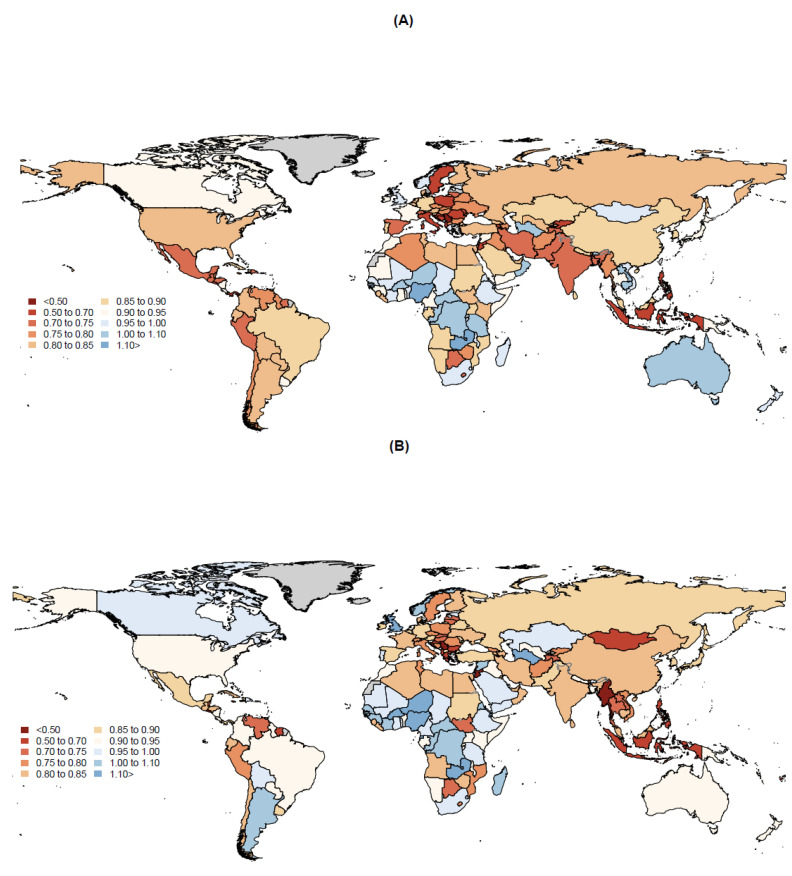
Geographic distribution of the observed to expected ratios of the impact of COVID-19 on tuberculosis diagnoses in 2020 (**A**) and 2021 (**B**). Countries that reported at least 50 all-form tuberculosis cases in 2019 were included in this analysis. Therefore, countries shaded in grey represent countries that were not included in the analysis.

**Figure 2 microorganisms-11-02191-f002:**
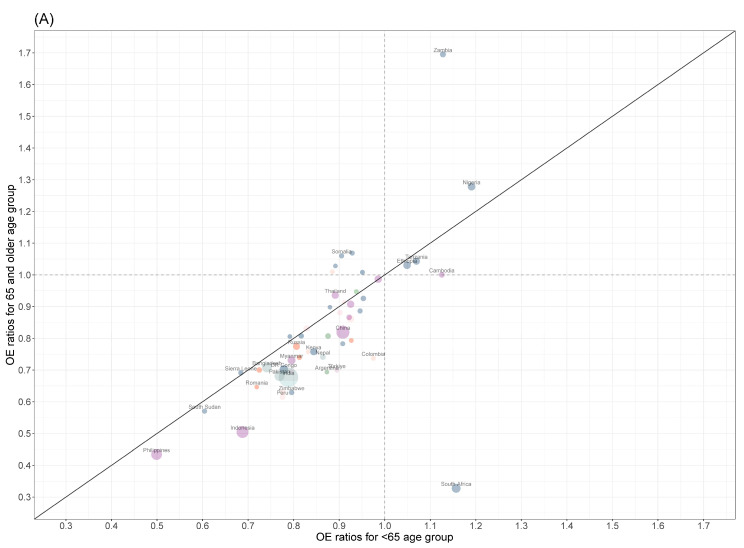
Observed-to-expected (OE) ratios of COVID-19 impact on tuberculosis diagnoses for the ≥65-year-old and under 65-year-old age groups in 2020 (**A**) and 2021 (**B**) for countries that reported at least 10,000 tuberculosis cases in 2019. The sizes of circles are weighted by country-specific contributions of global notified TB cases in 2019. Colors of circles represent Global Burden of Disease (GBD) super-regions.

**Figure 3 microorganisms-11-02191-f003:**
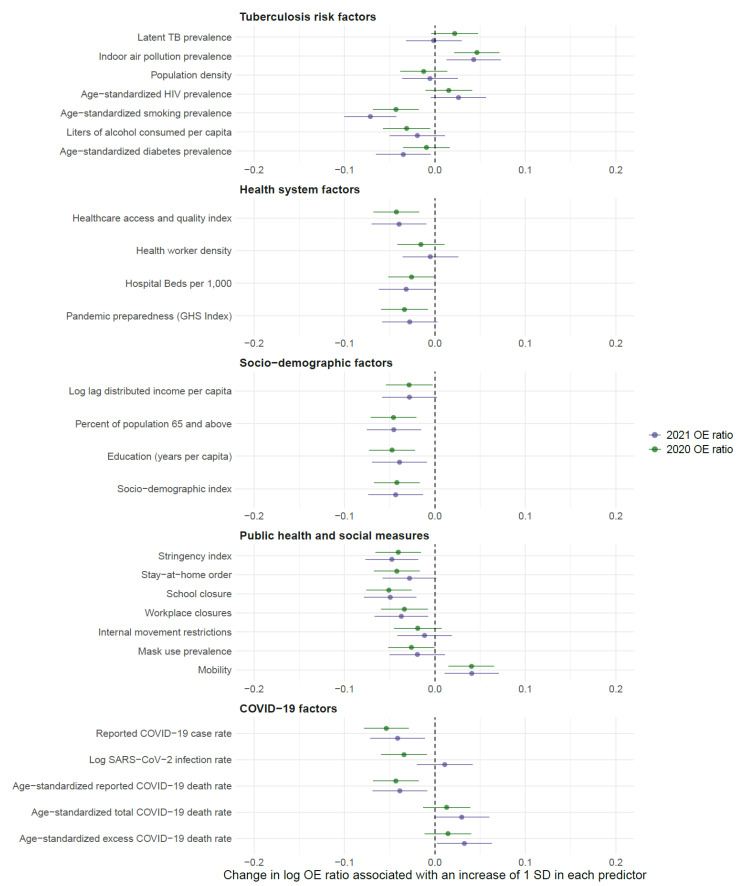
Bivariate associations between tuberculosis risk factors, health system, socio-demographic, public health and social measures, and COVID-19 factors on the observed-to-expected ratios of tuberculosis diagnoses. The points represent standardized coefficients, while the shaded lines represent the corresponding 95% confidence intervals of the coefficient. SD = standard deviation.

**Table 1 microorganisms-11-02191-t001:** Difference in observed-to-expected tuberculosis diagnoses during the COVID-19 pandemic by Global Burden of Disease super-regions in 2020 and 2021.

Location	Year	Observed TB Notifications		Expected TB Notifications		Difference in Observed to Expected TB Notifications	
		Cases	Rate per 100,000 Population	Cases	Rate per 100,000 Population	Number	Ratio
Global	2020	5,830,000	74.5	7,380,000 (7,090,000, 7,680,000)	94.4 (90.6, 98.2)	−1,550,000 (−1,850,000, −1,260,000)	0.79 (0.75, 0.83)
	2021	6,430,000	81.5	7,710,000 (7,330,000, 8,100,000)	97.7 (92.8, 102.6)	−1,280,000 (−1,670,000, −897,000)	0.83 (0.79, 0.88)
Southeast Asia, East Asia, and Oceania	2020	1,740,000	80.2	2,350,000 (2,240,000, 2,450,000)	107.8 (103.0, 112.7)	−601,000 (−706,000, −496,000)	0.74 (0.70, 0.78)
	2021	1,730,000	79.2	2,450,000 (2,310,000, 2,590,000)	112.2 (105.7, 118.6)	−721,000 (−863,000, −580,000)	0.71 (0.66, 0.75)
Central Europe, Eastern Europe, and Central Asia	2020	136,000	32.4	170,000 (165,000, 174,000)	40.6 (39.5, 41.6)	−34,200 (−38,600, −29,800)	0.80 (0.78, 0.82)
	2021	136,000	32.5	160,000 (155,000, 165,000)	38.3 (37.1, 39.5)	−24,100 (−29,000, −19,100)	0.85 (0.82, 0.88)
High-income	2020	84,600	7.8	98,200 (96,600, 99,700)	9.0 (8.9, 9.2)	−13,600 (−15,200, −11,900)	0.86 (0.85, 0.88)
	2021	85,900	7.9	95,100 (93,200, 96,900)	8.7 (8.5, 8.9)	−9140 (−11,000, −7230)	0.90 (0.88, 0.92)
Latin America and the Caribbean	2020	177,000	30.0	217,000 (212,000, 222,000)	36.8 (36.0, 37.6)	−40,000 (−45,000, −35,000)	0.82 (0.80, 0.84)
	2021	191,000	32.1	221,000 (215,000, 227,000)	37.1 (36.1, 38.2)	−29,900 (−36,000, −23,700)	0.86 (0.84, 0.89)
North Africa and the Middle East	2020	158,000	25.6	192,000 (187,000, 197,000)	31.2 (30.4, 32.0)	−34,100 (−39,000, −29,100)	0.82 (0.80, 0.85)
	2021	167,000	26.9	197,000 (191,000, 204,000)	31.7 (30.7, 32.7)	−29,900 (−36,200, −23,600)	0.85 (0.82, 0.88)
South Asia	2020	2,160,000	118.2	2,950,000 (2,670,000, 3,220,000)	161.3 (146.3, 176.3)	−787,000 (−1,060,000, −513,000)	0.73 (0.65, 0.81)
	2021	2,640,000	143.0	3,130,000 (2,780,000, 3,490,000)	169.7 (150.5, 189.0)	−493,000 (−849,000, −138,000)	0.84 (0.74, 0.95)
Sub-Saharan Africa	2020	1,370,000	123.9	1,410,000 (1,380,000, 1,450,000)	127.8 (124.8, 130.8)	−43,900 (−77,200, −10,600)	0.97 (0.95, 0.99)
	2021	1,480,000	130.6	1,450,000 (1,410,000, 1,500,000)	128.3 (124.6, 131.9)	26,100 (−15,500, 67,700)	1.02 (0.99, 1.05)

Countries included in each of the Global Burden of Disease super-regions are shown in Appendix A.

**Table 2 microorganisms-11-02191-t002:** Multivariable linear regression results for the observed-to-expected (OE) ratios of tuberculosis diagnoses during the COVID-19 pandemic stratified by year.

			Coefficient (95% CI)	Standardized Coefficient (95% CI)	Variation in OE Ratio Explained by Each Factor
**2020 Model**	Age-standardized smoking prevalence (per 5%)		0.973 (0.957, 0.990)	−0.044 (−0.072, −0.016)	16.4%
	Education (2 years per capita)		0.986 (0.964, 1.009)	−0.020 (−0.053, 0.012)	9.6%
	Percent of population 65 and above (per 5%)		0.971 (0.944, 0.999)	−0.038 (−0.074, −0.002)	12.0%
	Stay-at-home order (per 30 days in place)		0.993 (0.985, 1.000)	−0.025 (−0.051, 0.002)	12.3%
	School closure (per 30 days in place)		0.988 (0.977, 0.998)	−0.033 (−0.061, −0.006)	17.6%
	Reported COVID-19 case rate (per 1000)		1.001 (0.999, 1.003)	0.016 (−0.018, 0.051)	9.0%
	Log SARS-CoV-2 infection rate (per 25%)		0.991 (0.987, 0.996)	−0.052 (−0.080, −0.025)	16.6%
	Age-standardized reported COVID-19 death rate (per 1000)		0.984 (0.947, 1.022)	−0.011 (−0.037, 0.015)	6.5%
Model characteristics	R^2^	0.288			
	AIC	−614.373			
**2021 Model**	Age-standardized smoking prevalence (per 5%)		0.957 (0.940, 0.974)	−0.071 (−0.100, −0.043)	
Model characteristics	R^2^	0.121			
	AIC	−565.799			

## Data Availability

All data utilized in this analysis are publicly available at https://www.who.int/teams/global-tuberculosis-programme/data (accessed on 22 August 2023) for TB notifications from 2013 to 2021; https://github.com/OxCGRT/covid-policy-tracker (accessed on 4 June 2023) for public health and social measures; https://www.ghsindex.org/report-model/ (accessed on 4 June 2023) for global health security index; https://covid19.healthdata.org/global (accessed on 4 June 2023) for IHME COVID-19 data; and https://ghdx.healthdata.org/record/global-burden-disease-study-2019-gbd-2019-covariates-1980-2019 (accessed on 28 May 2023) for Global Burden of Disease 2019 covariate data.

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
