# Peer review of "Global-, Regional-, and National-Level Impacts of the COVID-19 Pandemic on Tuberculosis Diagnoses, 2020–2021"

_microorganisms, 2023, doi:10.3390/microorganisms11092191_

Round 1

Reviewer 1 Report

The manuscript  Global-, regional-, and national-level impacts of the COVID-19 2 pandemic on tuberculosis diagnoses, 2020-2021 , It deals with the investigation of the impact of the COVID-19 pandemic on TB cases in some countries of the world.

The subject addressed by the paper is of great interest, considering that TB continues to be a global health problem.

This manuscript addresses the impact of the COVID19 pandemic on the diagnosis of TB in several countries.

The experimental design is robust, the presenting the data is observed to be organized, and it is probably a good contribution to understanding the behavior of the pandemic in the health of patients diagnosed with TB.

The content of the work shows a good analysis of the impact of the actions carried out to control the pandemic, such as Stay-at-home orders, school closures, SARS-CoV-2 infection rates, and aging of the population were associated with an increased difference between observed TB diagnoses and those expected.

Author Response

Thank you for taking time to carefully review our analysis and for your positive feedback.

Reviewer 2 Report

The authors comprehensively examined the impact of the COVID-19 pandemic on tuberculosis (TB) detection across 170 nations. This comprehensive analysis involved a comparison between the actual TB diagnoses observed in 2020 and 2021 and the projected diagnoses in the absence of the pandemic. The influence of COVID-19 on TB diagnoses was investigated for two distinct age groups: individuals aged over 65 and those under 65. Furthermore, the authors set out to identify pivotal factors among the 27 pertinent variables associated with both the pandemic and TB. These factors were determined to be the primary contributors to the variance in COVID-19's effect on TB diagnoses across all age segments, spanning both the national and global levels.

The insights derived from these meticulous evaluations play an indispensable role in understanding the global repercussions of the pandemic on TB diagnoses and the nuances seen among diverse nations.

The study is notably well-structured, and its findings are presented with remarkable clarity. However, I do have some constructive feedback concerning the tables and figures included.

For instance, Table 1 appears excessively lengthy and could benefit from simplification to enhance readability and accessibility.

Furthermore, the figures, although informative, tend to be visually congested. Consider refining the presentation, particularly in the case of Figure 2. Limiting the display to key countries and employing a concise, declarative title to highlight the core message will significantly enhance the figure's legibility.

Lastly, it would be prudent to ensure that all figures are of optimal resolution, guaranteeing the highest quality presentation.

None.

Author Response

Thank you for your thoughtful and constructive comments. We have now simplified Table 1 so that only results for the 7 Global Burden of Disease (GBD) super-regions1 and global-level results are presented. Results for all 21 GBD regions and the 170 included countries can be found in Table S1 in the appendix. We also agree that Figure 2 could be improved to enhance clarity. We have opted to only show countries that reported at least 10,000 tuberculosis cases in 2019 while improving the axes, which has helped to improve clarity of the figure. We have also provided high-resolution figures in PDF format.

Reference

  1. Murray CJ, Ezzati M, Flaxman AD, Lim S, Lozano R, Michaud C, et al. GBD 2010: design, definitions, and metrics. The Lancet. 2012 Dec;380(9859):2063–6.